# Learning What to Remember: Long-term Episodic Memory Networks for Learning from Streaming Data

## Abstract

Current generation of memory-augmented neural networks has limited scalability as they cannot efficiently process data that are too large to fit in the external memory storage. One example of this is lifelong learning scenario where the model receives unlimited length of data stream as an input which contains vast majority of uninformative entries. We tackle this problem by proposing a memory network fit for long-term lifelong learning scenario, which we refer to as Long-term Episodic Memory Networks (LEMN), that features a RNN-based retention agent that learns to replace less important memory entries based on the *retention probability* generated on each entry that is *learned* to identify data instances of generic importance relative to other memory entries, as well as its historical importance. Such learning of retention agent allows our long-term episodic memory network to retain memory entries of generic importance for a given task. We validate our model on a path-finding task as well as synthetic and real question answering tasks, on which our model achieves significant improvements over the memory-augmented networks with rule-based memory scheduling as well as an RL-based baseline that does not consider relative or historical importance of the memory.

## 1 Introduction

In a real world machine learning problem, it is common to process a tremendous amount of sequential data (e.g. dialogues, videos, and medical records). To model such sequential data, researchers often use recurrent neural networks (RNNs) because of its capability to process sequentially dependent data. Recently, popular variants of RNNs such as Long Short-Term Memory (LSTM, Hochreiter & Schmidhuber (1997)) and Gated Recurrent Unit (GRU, Chung et al. (2014)) have achieved impressive performances on a number of tasks, such as machine translation, image annotation, and question answering. Nevertheless, despite such impressive achievements of the recurrent neural networks, it is still challenging to model long sequential data such as dialogues since RNNs have a limited memory capacity. The only memory that LSTM or RNN contains is a fixed dimensional vector. The LSTM or RNN performs memorization by adding to, and erasing, this single vector. This limited memory capacity can be problematic when dealing with a long sequence.

To overcome this scalability problem, researchers have proposed neural network architectures that contain external memory module, including Neural Turing machine (NTM, Graves et al. (2014)) and End-to-End Memory Networks (Sukhbaatar et al. (2015)). These memory-augmented networks have mechanisms to write representation to the memory cells and read from them. The addressing of the memory cells can be done either by an index to perform sequential read/write or by contents to perform a content-based access. The memory-augmented networks have been shown to be useful for memory-oriented tasks such as copying, pathfinding, and question answering. Yet, they are mostly targeting toy problems, using the external memory as a working memory in the human memory mechanism, and need to manage a relatively small number of memory entries. Thus, there has been less focus on how to efficiently manage the memory.

However, when the network needs to process data that are too large to fit into the memory, we need to carefully manage the memory cell such that the external memory module contains the most informative pieces of data. In other words, we should be able to determine which memory cell is

unlikely to be used in the future so it can be replaced with the incoming data when memory is full. To this end, we consider this memory management problem as a learning problem and train an agent by reinforcement learning, such that it learns to keep the most important information in memory and evict unimportant ones.

How can we then select which memory entries are most important? To achieve this goal, we propose to train the memory module itself using reinforcement learning to delete the most uninformative memory entry in order to maximize its reward on a future task. However, this is a challenging task since for most of the time, this scheduling should be performed without knowing which task will arrive when. Thus, deciding which memory to keep and which to erase should be done by considering relative generic importance of the memory entries. To tackle this challenge, we implemented the memory retention mechanism using a spatio-temporal recurrent neural network, that can learn relative importance among the memory entries as well as their historic importance. We refer to this memory-augmented networks with spatio-temporal retention mechanism as Long Term Episodic Memory Networks (LEMN). LEMN can perform selective memorization to keep a compact set of important pieces of data that will be useful for future tasks in lifelong learning scenarios. We validate our proposed retention mechanism against naive scheduling method as well as RL-based scheduling on three different tasks, namely path-finding, episodic question answering and long question answering, against which it significantly outperforms.

Our contribution is twofold:

- We consider a novel task of learning from streaming data, where the size of the memory is significantly smaller than the length of the data stream.
- To implement a retention mechanism, we propose a retention agent that can be integrated with existing external memory neural networks, which is trained with reinforcement learning to keep memory cells of general importance.

## 2 RELATED WORK

**Memory-augmented networks**   Graves et al. (2014) propose Neural Turing Machine (NTM), which has a memory and a controller that reads from and writes to memory. The controller composed of read heads and write heads uses soft-attention mechanism to access memory for differentiable memory access. The NTM has two addressing mechanisms: content-based addressing and location-based addressing. Location-based addressing allows a data sequence to be stored in consecutive memory cells to preserve its sequential order. However, once the write head accesses a distant memory cells, the sequential order of information in consecutive memory cells is no longer preserved. Graves et al. (2016) propose Differentiable Neural Computer that extends the NTM to address the issue by introducing a temporal link matrix. As it is costly to write into memory while preserving its sequential order, memory-augmented networks with write mechanism are mostly used in cases where it is not necessary to track which sequential order the memory has been written (Santoro et al., 2016; Vinyals et al., 2016; Kaiser et al., 2017; Kim et al., 2018). Unlike NTM, End-to-End Memory Networks (MemN2N) (Sukhbaatar et al., 2015) and Dynamic Memory Networks (DMN+) (Xiong et al., 2016) do not have write mechanism but store all the inputs into memory. For this reason, the sequential order of the inputs are preserved at no extra cost; thus they are more suitable for episodic question answering problems such as bAbI tasks (Weston et al., 2015). In addition, they have sophisticated read mechanisms that allow to reason through multiple hops (or passes in DMN+). Seo et al. (2016) propose more advanced soft-attention based read mechanism, Bi-Directional Attention Flow (BiDAF), which obtains impressive performance on a difficult reading comprehension dataset, Stanford Question Answering Dataset (SQuAD, Rajpurkar et al. (2016)). Oh et al. (2016) extend MemN2N to train deep reinforcement learning agents, Memory Q-Network (MQN) and Feedback Recurrent Memory Q-Network (FRMQN), to solve 3D Mazes. However, such neural networks have the limitation that the size of external memory should be large enough to store all the data.

**Memory Retention Policy**   Most of the existing approaches (MemN2N, DMN+, BiDAF and FRMQN) do not consider the case where memory becomes full, and simply truncate the sequence of data to the size of memory if it is longer than the memory size. Many of mutable external memory neural networks employ least-recently-used (LRU) based memory retention policy, which overwrites new data into the least used memory cell. This policy, although more reasonable than FIFO, is still

limited as it is a hand-crafted rule and does not consider actual long-term dependencies between a memory entry and the task. This is a critical limitation in lifelong learning setting, since some of the memory entries written in the long past and have not been accessed for long may still be necessary to respond to queries that arrive in the far future. Our work, on the other hand, is able to learn such long-term dependencies. DNTM (Gülçehre et al., 2016) learns where to overwrite using reinforcement learning as done in our work. Yet, it only considers the pairwise relationships between the current data instance and each individual memory entries, while our model learns both relative importance and historic importance of a memory entry using a spatio-temporal RNN architecture.

# 3 LEARNING WHAT TO REMEMBER FROM STREAMING DATA

We consider the problem of learning from a long data stream that contains large portion of unimportant, noisy data (e.g. routine greetings in dialogs) with limited memory. Formally, an agent $\mathcal{A}$ takes as input a streaming data (e.g. sentences or images) $X = \{x_1, \cdots, x_T\}$ and manages an external memory $M = [m_1, \cdots, m_N], m_i \in \mathbb{R}^d$ where $T \gg N$. Thus, the agent should decide which memory cell to evict to store incoming data. To this end, the agent should learn the relative importance of memory entries for a future task. In traditional reinforcement learning scheme, we can formulate this problem as learning the policy $\pi(a_t|s_t)$ to maximize a return $R$, where action $a_t$ is a memory cell to erase, and state $s_t = [M_t; x_t]$, where $M_t$ is the current memory and $x_t$ is the input at time $t$. The agent appends $x_t$ to the memory $M_t$ until it reaches the maximum size $N$.

If the memory is full, the agent erases one memory cell based on the policy $\pi(m_i|M_t, x_t)$ to append $x_t$. Thus, we can consider $1 - \pi(m_i|M_t, x_t)$ as the *retention* value of each memory. At arbitrary time step $t_f$, it encounters a task $\mathcal{T}$ (e.q. question answering) with a reward $R_{\mathcal{T}}$, whose reward is defined by the specific task. For QA task, the reward will be 1 if it generated a correct answer and $-1$ otherwise. The instance $x_t$ which arrive at timestep $t$ can be in any data format, and is transformed into a memory vector representation $e_t \in \mathbb{R}^d$ to be stored in memory.

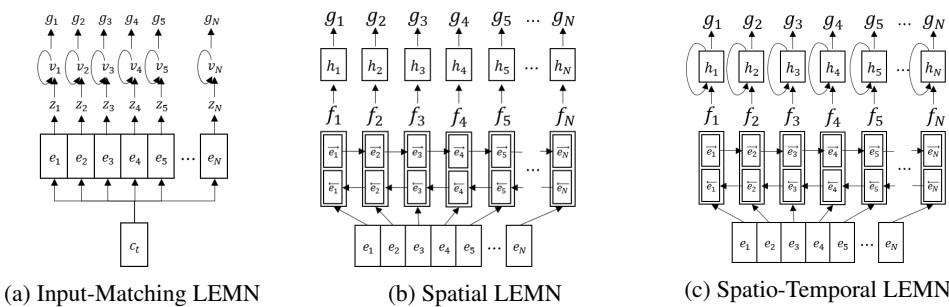

(a) Input-Matching LEMN      (b) Spatial LEMN      (c) Spatio-Temporal LEMN

Figure 1: Illustration of memory-retention architectures

## 3.1 MEMORY-RETENTION MECHANISMS

In this section, we describe three different memory-retention mechanisms in detail. We first encode input $x_t$ and each memory cell $m_{t,i}$ to a vector representation as follows:

$$c_t = \psi(x_t)$$
$$e_{t,i} = \varphi(m_{t,i})$$

where $c_t, e_{t,i} \in \mathbb{R}^d$, $\psi(\cdot)$ can be an input embedding similar to RNN controller in (Gülçehre et al., 2016) or position encoding in (Sukhbaatar et al., 2015) and $\varphi(\cdot)$ can be a memory encoding layer of the base external memory neural network. For example, MemN2N (Sukhbaatar et al., 2015) uses a bag of words and embedding matrices to convert to memory representation.

**Input-Matching LEMN** This is the simplest RL-based retention mechanism that is similar to DNTM from Gülçehre et al. (2016) and utilizes the memory usage information, which we refer to

as Input-Matching LEMN (IM-LEMN, Figure 1a). The usage of each memory representation $m_i$ is computed as the learned similarity between it and the current input $x_t$. As in Gülçehre et al. (2016), we least recently used (LRU) addressing by computing the exponential moving average $v_t$ of the logit $z_t$, the LRU factor $\gamma_t$, and the policy as follows:

$$v_{t,i} = 0.1v_{t-1,i} + 0.9z_{t,i}$$

$$\gamma_t = \sigma(W_\gamma c_t + b_\gamma)$$

$$g_{t,i} = z_{t,i} - \gamma_t v_{t-1,i}$$

$$\pi(m_i|M_t, x_t) = \text{softmax}(g_{t,i})$$

where $W_\gamma \in \mathbb{R}^d, b_\gamma \in \mathbb{R}$, softmax$(z_i)$ is $exp(z_i)/\sum_j exp(z_j)$, $\sigma(z)$ is $1/(1 + exp(z))$. This model estimates the policy based on the similarities between input $x_t$ and memory cells $m_{t,i}$.

**Spatial LEMN**   A major drawback of IM-LEMN is that the score of each memory depends only on the input $x_t$. In other words, the score is computed independently between memory cell and input but does not consider its relative importance to other data instances in the memory. To overcome this limitation, we propose Spatial LEMN (S-LEMN, Figure 1b) which computes the relative importance of memory cells to its neighbors and other memory cells using a bidirectional GRU as follows:

$$\overrightarrow{e}_{t,i} = GRU_{\theta_{fw}}(e_{t,i}, \overrightarrow{e}_{t,i-1})$$

$$\overleftarrow{e}_{t,i} = GRU_{\theta_{bw}}(e_{t,i}, \overleftarrow{e}_{t,i+1})$$

$$f_{t,i} = \text{ReLU}(W_f[\overrightarrow{e}_{t,i}, \overleftarrow{e}_{t,i}] + b_f)$$

where $W_f \in \mathbb{R}^{2d \times d}, b_f \in \mathbb{R}^d$, $GRU_\theta$ is a Gated Recurrent Unit parameterized by $\theta$, $[\overrightarrow{e}_{t,i}, \overleftarrow{e}_{t,i}]$ is a concatenation of features, ReLU is a rectified linear unit. We use multi-layer perceptron (MLP) with scalar output to estimate the policy as follows:

$$h_{t,i} = W_h f_{t,i} + b_h$$

$$g_{t,i} = W_g h_{t,i} + b_g \tag{1}$$

$$\pi(m_i|M_t, x_t) = \text{softmax}(g_{t,i})$$

where $W_h \in \mathbb{R}^{d \times d/4}, b_h \in \mathbb{R}^{d/4}, W_g \in \mathbb{R}^{d/4}, b_g \in \mathbb{R}$. Thus, it can compute the general importance of memory cell itself and the relation between its neighbors in contrast to IM-LEMN.

**Spatio-Temporal LEMN**   In episodic tasks, the importance of memory also changes over time and tasks. Hence, we propose Spatio-Temporal LEMN (ST-LEMN, Figure 1c) that uses a GRU over time to consider the historical importance as well as relative importance of each memory entry. We simply replace the hidden layer in Equation (1) with a GRU over time as follows:

$$h_{t,i} = GRU_{\theta_h}(f_{t,i}, h_{t-1,i})$$

$$g_{t,i} = W_g h_{t,i} + b_g$$

$$\pi(m_i|M_t, x_t) = \text{softmax}(g_{t,i})$$

**Memory Update**   Using aforementioned memory-retention mechanism, the agent samples the memory cell index $i$ from the probability distribution $\pi(m_i|M_t, x_t)$. Then it erases the $i_{th}$ memory cell and appends the input $x_t$ as follows:

$$M_{t+1} = [m_1, \cdots, m_{i-1}, m_{i+1}, \cdots, m_N, x_t].$$

**No Operation (NOP)** As done in Gülçehre et al. (2016), for IM-LEMN we add a NOP memory cell at the end of memory to consider the case where the input is not written to the memory. For S-LEMN and ST-LEMN, we append $x_t$ to the place of NOP such that it could be selected for removal.

We integrate these retention mechanisms into base memory-augmented neural networks to enable to efficiently maintain its external memory. The details of the base networks are given in the Experiment section. We use Asynchronous Advantage Actor-Critic (A3C, Mnih et al. (2016)) with Generalized Advantage Estimation (GAE, Schulman et al. (2015)) to train all models.

## 4 EXPERIMENT

We evaluate the proposed retention agent on three different tasks in following subsections.

### 4.1 MAZE

Oh et al. (2016) proposed a task for memory-based deep reinforcement learning (RL) agents, where the agent should navigate through a 3D maze and enter the correct exit, and demonstrated that the agents with external memory outperform the agents without external memory. However, they assumed that the agent had a sufficiently large memory to store every observed cell, although the actual amount of information was not as much as the size of the memory. In this experiment, we compare the navigator agent with and without ST-LEMN. We followed the same experimental setup as used in Oh et al. (2016) except that we use A3C algorithm (Mnih et al. (2016)) with GAE (Schulman et al. (2015)) instead of Q-Learning algorithm (Mnih et al. (2015)) to train the agent, and used only three actions - Move forward, Look Left, Look Right - for expedited training. We use MQN and FRMQN as base networks to see the effect of the model without recurrent connections and with recurrent connections. We compared the performance of these base models with learned retention and FIFO scheduling under two different environments.

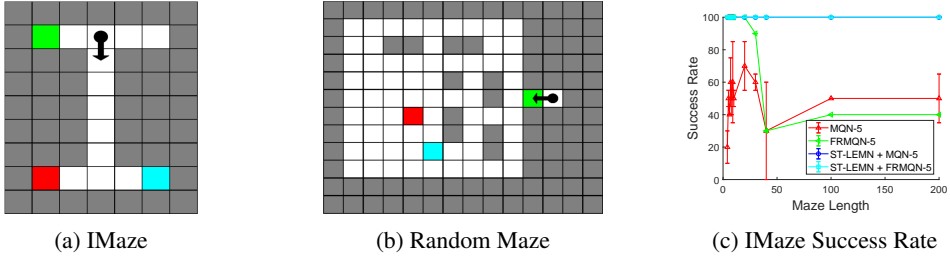

|          (a) IMaze          |       (b) Random Maze       |    (c) IMaze Success Rate    |

Figure 2: (a) Example of IMaze environment. (b) Example of Random Maze environment. (c) Success rate of memory-based agents using FIFO memory scheduling and our retention mechanism.

**IMaze** This environment contains an I-shaped maze, where the agent should reach the correct exit based on the color of the initial cell (See Figure 2a). We train the models on mazes with varying lengths of corridors $N = \{5, 7, 9\}$, and validate on mazes with corridors with 15 different lengths $N = \{4, 6, 8, 10, 15, 20, 25, 30, 35, 40, 100, 200\}$. We set the memory size to 4 for all compared models. We observe that the agent with ST-LEMN successfully retains the indicator information in its memory while passing through the long corridor by removing useless frames that describe the corridor (Figure 3a and 3b). At the end of corridor, agent with ST-LEMN retrieves stored indicator frame to decide which way to go. Agents with FIFO scheduling do not retain the indicator information and always exits at the same goal. FRMQN, which is an agent that has recurrent connection between time in its context embedding can complete task correctly on mazes with short corridor even with FIFO scheduling. Yet, it fails on mazes with long corridors since it is difficult to learn long-term dependencies without explicitly storing the cell in the memory. In contrast, we observed that the agents with ST-LEMN can solve the maze with small fixed-sized memories regardless of the length of maze, by learning the long-term importance of input data instances (Figure 2c).

**Random Maze: Single Goal with Indicator** We also test with the randomly generated maze as in Oh et al. (2016) (Figure 2b), testing for the Single Goal with Indicator (SingInd) task, where the agent should reach the correct goal based on the indicator that can be observed at the starting

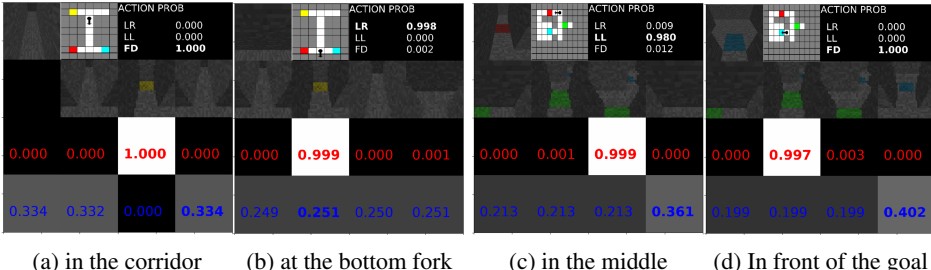

| (a) in the corridor | (b) at the bottom fork | (c) in the middle | (d) In front of the goal |

Figure 3: Visualization of MQN retention agent's status: (a-b) I-Maze and (c-d) Single Goal with Indicator. The first row shows a current view of an agent, a top-down view of the maze with the current position of the agent, and action probability distribution. Each column from the second row shows the content of memory cells. The third and the fourth row indicate the attention value and the drop probability of each memory cell.

position. As shown in Table 1 the retention agent has significantly higher success rate compared to the agent with FIFO scheduling, as it retains the indicator frame while it navigates through the random maze. As shown in 3c, 3d the retention agent retains the indicator discovered at the start of maze in its memory and retrieves its information when it needs to make a decision.

Table 1: Best success rates on SingleInd Task.

| Task | Test | Large |
|---|---|---|
| FIFO | 68.90 | 60.40 |
| ST-LEMN | **73.20** | **74.80** |

(a) MQN-5

| Task | Test | Large |
|---|---|---|
| FIFO | 84.30 | 91.20 |
| ST-LEMN | **88.50** | **93.70** |

(b) FRMQN-5

## 4.2 SYNTHETIC EPISODIC QUESTION ANSWERING

**Dataset** Weston et al. (2015) created a synthetic dataset for episodic question answering, called bAbI, that consists of 20 tasks. Among tasks, we evaluate memory-retention mechanisms on two supporting facts task (task 2, Figure 4a). Additionally, We generate noisy and large version of the two supporting facts task from open-sourced template Weston et al. (2015). Each task equally has five questions in an episode, where all questions share context sentences given in the episode. In the noisy two supporting fact task, which we refer to as **Noisy** task (Figure 4b), we inject noise sentences into original two supporting facts task to investigate the ability of retention mechanism to filter out the noise. We organize this synthetic dataset as follows: 60% of dataset have no noise sentence; 10% of dataset have approximately 30% noise sentences; 10% of dataset have approximately 45% noise sentences; 10% of dataset have approximately 60% noise sentences. Totally, the length of each episode is fixed to 45 (5 questions + 40 facts). Questions are placed after every 8 facts. In large and noisy two supporting facts task, which is called **Large** task (Figure 4c), composition of tasks is similar to Noisy but the length of episode and the position of questions vary. The length of each episode is randomly chosen between 20 and 80. Questions can be placed anywhere in episode.

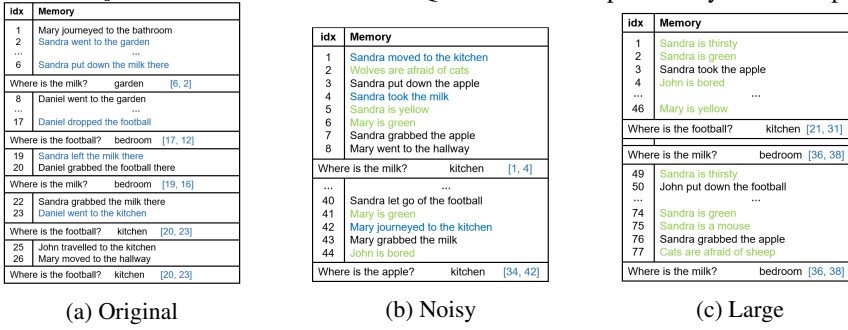

| (a) Original | (b) Noisy | (c) Large |

Figure 4: Example of (a) Original task, (b) Noisy task and (c) Large task. Sentences in green are noise sentences and ones in blue are supporting facts of each question.

**Training Detail** We use MemN2N(Sukhbaatar et al. (2015)) as base networks. Particularly, we use base MemN2N with position encoding representation, 3 hops and adjacent weight tying. We set

the dimension of memory representations to $d = 20$. We compare FIFO, IM-LEMN, S-LEMN and ST-LEMN on the two supporting facts task, and two modified tasks. IM-LEMN on this task uses an average of attention logits from each hop as $z_t$ in Equation (3.1). Also, it embeds input using GRU similar to a GRU controller in Gülçehre et al. (2016). S-LEMN and ST-LEMN use the value memory of the first hop in the base MemN2N as a memory representation $e_{t,i}$ in Equation (3.1). We constrain the size of memory as 5 or 10 to validate the scheduling performance of our retention mechanism. Since we jointly train the agent for QA task and memory retention, for stable training we pretrain MemN2N with FIFO mechanism for 50k steps and then go on with training other mechanisms. We train our models using ADAM optimizer (Kingma & Ba (2014)) with the learning rate of 0.001 for 200k steps on the Original and Noisy tasks, and for 400k steps on the Large task.

**Results** Table 2 shows the results on synthetic episodic question answering tasks. Sukhbaatar et al. (2015) reports the lowest error by multiple agents to cope with large variance in performance, and we follow this evaluation measure. For the Original and Noisy tasks, we measure error rate using best performing parameter among three takes of training, and among five takes for the Large task. As shown in table 2, our suggested memory retention mechanism significantly outperforms naive policy and policy inspired by LRU in all cases. For more detailed analysis, we present two sampled data from IM-LEMN and ST-LEMN in Figure 5. IM-LEMN's selection policy seems to largely depend on the current context, and therefore it deletes not only noise sentences but also informative sentences as well. Compared to IM-LEMN, memory of ST-LEMN does not have any noise sentences in its memory but evenly contains informative facts about location and object acquisition.

Table 2: Best error rates on bAbI tasks

| Task | Original | Noisy |
|---|---|---|
| FIFO | 41.40 | 50.20 |
| IM-LEMN | 23.70 | 40.20 |
| S-LEMN | 20.30 | 40.60 |
| ST-LEMN | **17.50** | **11.70** |

(a) Memory size 5

| Task | Original | Noisy | Large |
|---|---|---|---|
| FIFO | 16.50 | 44.10 | 32.40 |
| IM-LEMN | 16.10 | 18.90 | 9.00 |
| S-LEMN | 5.00 | 4.80 | **5.10** |
| ST-LEMN | **4.60** | **3.90** | 5.60 |

(b) Memory size 10

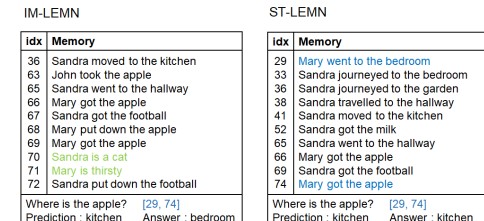

Figure 5: Examples of memory state of IM-LEMN and ST-LEMN on the Large task. Idx denotes index of sentence in one episode. Sentences in blue are supporting facts, and ones in green are irrelevant ones.

## 4.3 TRIVIAQA

**Dataset** TriviaQA(Joshi et al., 2017) is a question-answering dataset over 950K question-answer pairs in 662K evidence documents collected manually. The problem is difficult to solve by conventional models as it requires multiple reasoning with large number of sentences. The average number of words per document is 2895, which is infeasible to handle using existing reading comprehension models. We limit the number of words per document to 800 and use only questions that could be spanned within 400 words out of the 800 for training for expedited training. Also, we only use the first spanned word as labels since TriviaQA provides only the answer word and not their correct indices in the document. For a test set, we use all words in a document for each question-answer pair. Since the dataset does not provide labels for the test set, we use a validation set for test which contains both distant supervision set whose labels are collected automatically and verified set evaluated by the annotator. We evaluate our work only on the Wikipedia dataset, since previous work (Joshi et al., 2017), (Pan et al., 2017), (Yu et al., 2018) report similar results on both datasets.

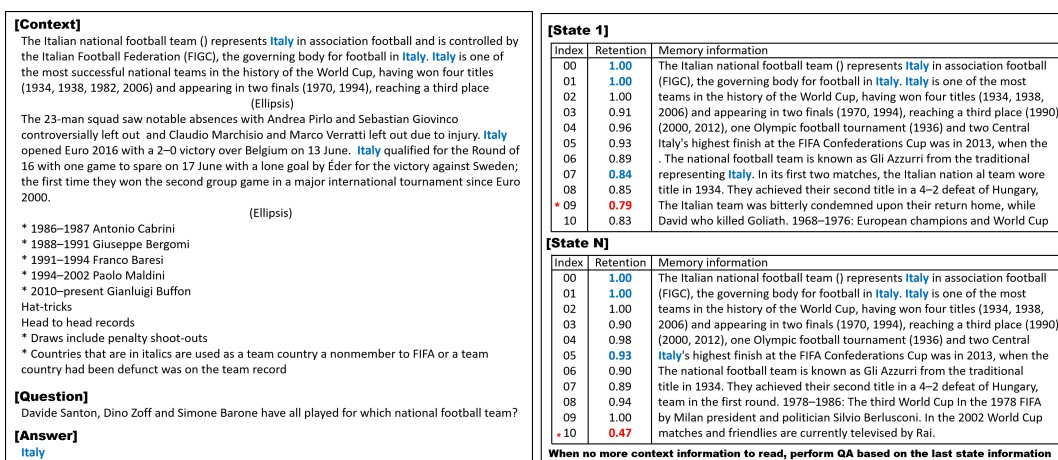

(a) Question-answering pair and its context      (b) The operation process of our model

Figure 6: **(a)** An example of TriviaQA dataset, **(b)** A visualization of how our model operates in order to solve the problem. LEMN sequentially reads the sentence one by one while scheduling the memory to retain only the important sentences. During the process in **(b)**, it retained the word 'Italy'(Thick blue) in order to answer a given question, by generating high retention value(Thick blue) on the memory cell containing 'Italy', and decides to erase uninformative memory cells with low retention value (Thick red / * Mark).

**Base network** As base network, we use BiDAF(Seo et al., 2016), which is one of the state-of-the-art reading comprehension model that predicts the indices for the exact location of the answer in the given document and modified the word-level intermediate representations to sentence-level representations using RNN to handle an entire sentence at a time. We set the memory size $N = 10$, and the word length per memory slot to 20. We trained our memory-augmented BiDAF using ADAM optimizer (Kingma & Ba, 2014) with the initial learning rate of 0.0001.

**Results** Table 3 shows the results on TriviaQA dataset. In (Joshi et al., 2017), they selected the highest score for ExactMatch and F1 score among multiple documents that could find the answer to a question. Our model outperforms all baselines because it has the ability to correctly identify informative sentences that are required to answer a given question. As shown in Figure 6, when new context information arrives at the model, our model determines which memory information is the most unnecessary based on the predicted the retention value.

Table 3: Q&A accuracy on Distant supervision **(Left)** and Verified **(Right)** subsets of Trivia dataset.

| Model | ExactMatch | F1score | | Model | ExactMatch | F1score |
|---|---|---|---|---|---|---|
| FIFO | 18.49 | 20.33 | | FIFO | 16.04 | 17.63 |
| IM-LEMN | 34.92 | 38.72 | | IM-LEMN | 32.07 | 33.75 |
| S-LEMN | 42.97 | 46.61 | | S-LEMN | 38.36 | 41.27 |
| ST-LEMN | **45.21** | **49.04** | | ST-LEMN | **44.33** | **47.24** |

## 5 CONCLUSION

We considered the problem of learning from streaming data, where the size of the data is too large to fit into the memory of a memory-augmented network. To solve the problem of retaining important data instances, we proposed Long-term Episodic Memory Network (LEMN), which is able to remember data instances of long-term generic importance. Using reinforcement learning, LEMN learns to decide which memory entry to replace when the memory becomes full, based on both relative importance between memory entries and their historical importance. We validated our LEMN on three different tasks, namely path finding, episodic question answering and long question answering against rule-based memory scheduling methods as well as an RL-agent trained without consideration of relative and historic importance of memory entries, against which it significantly outperforms. Further analysis of LEMN shows that such good performance comes from its ability to retain instances of long-term importance. As future work, we plan to apply our model to dialogue generation task for conversational agents.

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

## A  SHUFFLING MEMORY ENTRIES

We evaluate the effect of perturbing a sequential order of memory cells on an episodic question answering dataset. We perturb the sequential order of the memory of our models (S-LEMN and ST-LEMN) when estimating the policy $\pi(a_t|s_t)$ in training phase. We compare the models with the memory shuffled (Shuffled S-LEMN and Shuffled ST-LEMN) and ones with the sequential order preserved (S-LEMN and ST-LEMN) in Table 4. We observe that perturbing a sequential order results in the performance degeneration. This result shows that our models learn the relation between ordered memory cells to perform the episodic task where a sequential order is obviously important.

Table 4: Best error rates on bAbI tasks.

| Task | Original | Noisy | Task | Original | Noisy | Large |
|---|---|---|---|---|---|---|
| Shuffled S-LEMN | 37.50 | 41.60 | Shuffled S-LEMN | 12.70 | 38.40 | 17.70 |
| Shuffled ST-LEMN | 34.70 | 40.60 | Shuffled ST-LEMN | 7.10 | 38.00 | 6.30 |
| S-LEMN | 20.30 | 40.60 | S-LEMN | 5.00 | 4.80 | **5.10** |
| ST-LEMN | **17.50** | **11.70** | ST-LEMN | **4.60** | **3.90** | 5.60 |
| (a) Memory size 5 | | | (b) Memory size 10 | | | |

