# OpenReview forum: "Learning What to Remember: Long-term Episodic Memory Networks for Learning from Streaming Data"
_ICLR.cc/2019/Conference_

### Official Review · AnonReviewer1 · 2018-10-18
**Interesting idea and important problem area, but needs to be stress-tested**

**Rating:** 4
**Confidence:** 4

**Review:**

This work tackles the problems encountered by bounded memory storage mechanisms when faced with abundant data, of which much may be irrelevant or redundant. Such a problem is faced in lifelong learning settings, where a limitless data stream must somehow be encoded and stored so as to be useful at later points in time.

The researchers propose a solution based on “learning what to remember”. That is, rather than encode every observation (which can quickly become problematic), the model learns to replace less important memories. The importance of a memory is determined by its correlation with future reward; a “memory retention policy” is learned via reinforcement learning, wherein the model learns to retain or discard memories based on these actions’ (i.e., retentions) impact on future reward. Experiments to show the effectiveness of this mechanism include gridworld IMaze and Random Mazes, bAbI question answering (task 2), and Trivia QA.

Altogether the work does well to clearly describe an interesting approach to an important problem. The model is motivated and explained well, and there were no issues with understanding its inner workings.

Regarding the work’s novelty, there is a precedent for using RL-based write schemes (DNTM from Gulcehre et al, 2016), which the authors point out. I am not entirely convinced that the proposed writing scheme is a substantial addition over this past work, but I am not overly concerned about this since proper due credit is assigned in the paper. Perhaps a bit more discussion about the advantages of the proposed writing scheme could go a long way, since as it stands now, the paper simply claims that this past work “only considers the pairwise relationships between the current data instance and each individual memory”, and I’m not sure how much substance actually underlies this difference.

Unfortunately I think there is a fundamental problem with the work. The model is a proposed solution for problems with vast amounts of streaming data; problems that, presumably, current memory models would struggle with. However, the tasks in the paper do not fall in this domain. Instead, the authors chose to artificially cripple the size of their memory (using, for example, just a handful of memory “slots”) and demonstrate its performance on tasks that are otherwise completely within the realm of being solved by conventional memory models. This is fine as a jumping off point for the research, but for the model to be taken seriously as a valid solution to problems involving such a scale of data that current models cannot even cope, then it needs to show its worth on problems involving such a scale of data that current models cannot cope.

Demonstrating success here is important for a few reasons. First, such high-data scenarios may involve situations where many, many memories need to be encoded and considered for the future, since they are all useful or necessary for future performance. The experiments do not show whether the model can scale to, say, 100 or 1000 memories, which is within the realm of being “reasonable” for current memory architectures. Second, high-data scenarios may involve an abundant amount of distracting, irrelevant data. This places particularly tough demands on the RL-based writing mechanism, which will undoubtedly face problems with temporal credit assignment if: (a) the time between encoding and retrieval is long, and (b) there is high reward noise in the intermediate time. Thus, the authors should stress-test the components of their model, since these stresses will undoubtedly exist in the problems that the model is proposed to solve.

Some other minor considerations include the following. (1) The use of a single bAbI task is questionable. Why not run the model on the full suite? (2) How do conventional memory models perform on the tasks? Why are the baselines only variants of the proposed model?

To conclude and summarize, as a proposed solution to scenarios with streams of abundant data -- which the authors claim is a domain that current memory models may struggle -- the proposed model should tackle problems that: 1) have characteristics more reminiscent of these scenarios, and 2) are problems on which current memory models struggle, for the reasons claimed in the paper. In particular, it would be valuable to see model performance on tasks wherein very long stretches of time need to be considered. This is important because it can address questions with memory scaling (how does the model cope with more than a handful of memories?), and issues that would crop up in a reinforcement learning-based approach to memory retention over long time intervals (namely, long-term temporal credit assignment).

---

> ### Author Response · Authors · 2018-11-26
> **Response to AnonReviewer1**
>
> Thank you for your thoughtful comments regarding our paper.
>
> “Regarding the work’s novelty, there is a precedent for using RL-based write schemes (DNTM from Gulcehre et al, 2016), which the authors point out. I am not entirely convinced that the proposed writing scheme is a substantial addition over this past work, but I am not overly concerned about this since proper due credit is assigned in the paper.”
>
> As described in Section 3, IM-LEMN, a counterpart of DNTM, takes into account only matching scores between the current input and each memory cell to estimate the importance. Our addition to the model is two folds: spatial and temporal relation between memory cells. In Section 4.1 and 4.3, we have shown that our addition results in substantial performance improvements. In addition, to the best of our knowledge, not only does our work’s novelty involve proposing new writing scheme but it also involves posing the memory efficiency problem in lifelong learning and investigating how our models and the conventional models perform in the setting.
>
> “Instead, the authors chose to artificially cripple the size of their memory (using, for example, just a handful of memory “slots”) and demonstrate its performance on tasks that are otherwise completely within the realm of being solved by conventional memory models.”
>
> As described in Section 4.3, TriviaQA is a huge dataset so that conventional models such as BiDAF are not able to encode the entire context into memory due to the limitation of computation and memory cost of hardware. So, it is not that we cripple the size of our memory but we set the feasible size. In Section 4.1, I-Maze experiment, we show that conventional models (MQN and FRMQN) with our retention agent can find the appropriate goal regardless of the size of the maze. In contrast to our models, the conventional memory models require the memory as large as the maze.
>
> “(1) The use of a single bAbI task is questionable. Why not run the model on the full suite? (2) How do conventional memory models perform on the tasks?”
>
> We agree with that running the model on the full suite of bAbI tasks is of great interest. However, we focused more on running the TriviaQA experiments because bAbI tasks are synthetic but TriviaQA is a real question answering dataset.

---

### Official Review · AnonReviewer2 · 2018-10-28
**An interesting topic but need to think of strategy that is more reasonable to compute the similarity between each memory entry**

**Rating:** 4
**Confidence:** 4

**Review:**

This paper attempts to study memory-augmented neural networks when the size of the data is too large. The solution is to maintain a fix-sized episodic memory to remember the important data instances and at the same time erase the unimportant instances. To do so, the authors improve the method called DNTM (Gulcehre et al., 2016) by incorporating the similarity between each memory entry besides the similarity between the current data the each memory entry. Experiments show the effectiveness of the proposed method.

Here are my detailed comments:
This is an interesting topic where augmented memory is used to improve the performance of neural networks. It is important to put the most important information in the limited external memory and discard the less important contents. In the work DNTM, the similarity of the current data instance and each memory entry is introduced to determine which memory entry should be rewritten. The authors think that this measurement is not enough and consider the relationship between each memory entry. In my opinion, this is a reasonable extra measurement since the information is also important if it has strong connection with other stored information.

However, a deficiency of this work is that the relationship between each memory entry is not calculated in a reasonable way because the authors only use the bidirectional GRU to do this. From the motivation, we know that the authors want to obtain the relationship between every memory entry. However, as we know RNN models including GRU are suitable for those data that have sequence order. More specifically, bidirectional RNN models are used when we want to obtain not only the impact from beginning to end but also the impact from the end to the beginning. In addition, by using bidirectional RNN, we cannot obtain the relationship between each memory entry. If the authors want to realize that, it is necessary to disrupt the order of the memory entries and input the disordered entries into RNN models for n! times where n is the number of the memory entries and this will cost many computations. Although in experiments the proposed method shows its effectiveness and outperforms the baseline methods, the baseline methods are not enough to convince me that the proposed method is effective. I strongly suggest that the authors could incorporate more works that is state-of-the-art as baseline methods and consider strategies that are more reasonable to compute the relationship between each memory entry.

Besides, there are some grammar mistakes and typos, especially about the usage of article and correctness on singular and plural. The paper needs more careful proofreading.

---

> ### Author Response · Authors · 2018-11-26
> **Response to AnonReviewer2**
>
> Thank you for your thoughtful comments regarding our paper.
>
> “However, as we know RNN models including GRU are suitable for those data that have sequence order. More specifically, bidirectional RNN models are used when we want to obtain not only the impact from beginning to end but also the impact from the end to the beginning. In addition, by using bidirectional RNN, we cannot obtain the relationship between each memory entry. If the authors want to realize that, it is necessary to disrupt the order of the memory entries and input the disordered entries into RNN models for n! times where n is the number of the memory entries and this will cost many computations.”
>
> We thank the reviewer for kindly describing the way of computing the relation without the sequential order and its cost. However, incorporating the sequential order is our intention because the sequential order matters when estimating the importance of episodic memory cells. For example, in bAbI task 2, there are two sentences, “A went to X” and “A went to Y” in memory. Then, the importance of the latter should be greater than the former because the place where A was in is no longer useful to answer the future questions. Thus, we have modeled our writing mechanisms using a bidirectional RNN among the memory cells so that it can consider the spatial information of the memory. We added the experiment for the effect of disrupting the order of the memory in the revision in a similar way to [Santoro et al. 16] and observed that it results in the performance degeneration. For example, the error rate of ST-LEMN with memory size 5 on Noisy task increases by 28.9% point (see Table 4 in Appendix A) with occasional shuffling of memory entries.
>
> [Santoro et al. 16] Meta-Learning with Memory-Augmented Neural Networks, ICML 2016

---

### Official Review · AnonReviewer3 · 2018-11-02
**Important problem, interesting solutions but less convincing evaluation**

**Rating:** 5
**Confidence:** 5

**Review:**

Summary
========
The paper focuses on memory management problem of memory-augmented neural networks when the length of the streaming data is much larger than the number of memory entries. The paper proposes Long-term Episodic Memory Networks (LEMN) which learn a RNN-based agent to erase less important memory entries for storing incoming data by computing a retention score for each memory entry based on:
* The importance relative to other memory entries: a RNN through all memory entries.
* An entry’s historical importance: a RNN on an entry’s hidden values over time.

Comment
========
The target problem of memory management in MANN is of importance, and the solutions are interesting, especially the Spatio-Temporal LEMN, where both spatial dependencies between memory slots and temporal evolution of each slot itself are modeled.

However, the experiments give only proof of concepts without comparison against state-of-the-art for each task. For example, the paper lacks comparison with differentiable neural computer (DNC) [1], the well-known memory-augmented neural networks. Since the DNC also has the ability to keep track on the usage information of memory entries and decide whether to free them or not, there should be a comparison between the proposed LEMN and the DNC.

The model can be considered as an extension of the DNTM [2], referred to as IM-LEMN in the paper, with the introduction of recurrent connection over space and time. Although comparisons between the LEMN and IM-LEMN are available in section 4.2 and 4.3, there should be a similar comparison in section 4.1 to see whether the addition of recurrent connections brings benefits or not.

Abbreviations should be made clear. E.g., MQN should be written in the full form before using it. The MQN should be cited with Oh et al (2016).

References:

[1] Alex Graves, Greg Wayne, Malcolm Reynolds, Tim Harley, Ivo Danihelka, Agnieszka GrabskaBarwinska, Sergio Gomez Colmenarejo, Edward Grefenstette, Tiago Ramalho, John Agapiou, Adria Puigdomenech Badia, Karl Moritz Hermann, Yori Zwols, Georg Ostrovski, Adam Cain, Helen King, Christopher Summerfield, Phil Blunsom, Koray Kavukcuoglu, and Demis Hassabis. Hybrid computing using a neural network with dynamic external memory. Nature, 538(7626): 471–476, 2016. doi: 10.1038/nature20101.

[2] Caglar Gulc¸ehre, Sarath Chandar, Kyunghyun Cho, and Yoshua Bengio. Dynamic neural Turing machine with soft and hard addressing schemes. CoRR, abs/1607.00036, 2016.

---

> ### Author Response · Authors · 2018-11-26
> **Response to AnonReviewer3**
>
> Thank you for your thoughtful comments regarding our paper.
>
> “For example, the paper lacks comparison with differentiable neural computer (DNC) [1], the well-known memory-augmented neural networks.”
>
> We agree that the comparison with DNC is of great interest. However, we did not compare DNC with our model because it takes as an input a sequence of words instead of sentences, which makes the direct comparison difficult in terms of the memory size and its efficiency. Actually, this is the reason why we adopted the writing mechanism of DNTM, which is a counterpart of DNC, as our baseline.
>
> “Abbreviations should be made clear. E.g., MQN should be written in the full form before using it. The MQN should be cited with Oh et al (2016).”
>
> Thanks for your suggestion. In the revision, we clarified the full name of MQN and cited it in Section 2.

---

### Meta-Review · Area_Chair1 · 2018-11-05
**Important problem but the evalution is not good enough**

**Confidence:** 4
**Recommendation:** Reject

**Metareview:**


Pros:
- This is an interesting and relevant topic
- It is well motivated and mostly clear

Cons:
- The motivation, large amounts of data such as occur in lifelong learning, is not well examined in the evaluation which focuses on quite small problems.  For an example of work which addresses the lifelong memory management issue (though does not learn a memory management policy) see [1].
- In general the evaluation is not adequate to the claims.
- Reviewer 2 is concerned with the use of a bi-directional RNN for the comparison of memory entries since it may overfit to order.
- Reviewer 1 is somewhat concerned with novelty over other memory management schemes.

[1] Scalable Recollections for Continual Lifelong Learning. https://arxiv.org/pdf/1711.06761.pdf